Sign language use in healthcare: professionals’ insight

Qadhi Omaimah oqadhi@ksu.edu.sa
Medical-Surgical Nursing Department, College of Nursing, King Saud University , Riyadh , Saudi Arabia
Zhang Xin
Electronic publication date: 2025 Jun 2
Publication date: 2025
Volume: 13
Electronic Location ID: e19446
Received 2024 Sep 9; Accepted 2025 Apr 18
Copyright: ©2025 Qadhi
Copyright year: 2025
Copyright holder: Qadhi
License: This is an open access article distributed under the terms of the Creative Commons Attribution License, which permits unrestricted use, distribution, reproduction and adaptation in any medium and for any purpose provided that it is properly attributed. For attribution, the original author(s), title, publication source (PeerJ) and either DOI or URL of the article must be cited.
License URL: https://creativecommons.org/licenses/by/4.0/

Keywords: Sign language, Nursing, Quality of care delivery, Communication, Workplace, Deaf, Hearing-impaired, Healthcare providers, Special needs

Funding: The author received no funding for this work.

==============================
Background and Aim

Communication using sign language (SL) between health care providers (HCPs) and deaf and/or hearing-impaired (DHI) patients was reported to be difficult and oftentimes results in a compromised delivery of quality health care to patients. This study surveyed Saudi health care providers on their perception of SL knowledge on the provision of high-quality care to DHI patients.

Methods

This was a cross-sectional descriptive study among HCPs in different health and primary care centers in Riyadh, Saudi Arabia. The questionnaire was distributed officially by the Department of Surveys of King Saud University to target HCPs via email and through WhatsApp.

Results

A total of 238 HCPs were included in the study, of whom 180 (75.6%) were nursing professionals and 58 were from other health specialties. Only 15 (6.3%) of HCPs claimed to have received formal training in SL. Majority of the HCPs (n = 165, 69.3%) perceived that knowledge in SL is very important for communication with DHI patients and their families, whereas 65 (27.3%) perceived SL as somewhat important. Nurses believed that the quality of health service and care to deaf and/or hearing-impaired (DHI) patients is impacted by the inability of HCPs to communicate effectively and deliver high-quality care without the knowledge. Nursing professionals believed that knowledge of SL will improve the quality of care provided to DHI patients compared to other HCPs (97.2% vs. 87.9%, p = 0.005).

Conclusion

In order to provide DHI patients with high-quality healthcare, nurses believe that understanding SL is essential. Few nurses received formal training and few have adequate knowledge in SL. There is a need to provide nurses and HCPs adequate training in SL to improve communication with DHI patients and enhance DHI inclusivity in their management in line with the World Health Organization’s Universal Health Coverage and “health care for all”.

Introduction

People who are deaf and who have hearing impairment (DHI) often encounter challenges and their difficulty in hearing or speaking can be frustrating and isolating (Pullagura et al., 2024). These individuals usually have fewer educational and job opportunities, socially withdrawn due to reduced access to services and have emotional problems due to lack of self-esteem and confidence (Scheier, 2009). DHI faced a lack of accessible resources for end-of-life care, limiting their ability to understand and implement advance care planning (Cerilli et al., 2023). In Saudi Arabia, the stated prevalence rate of hearing disability is at 3.3% and speech disability is at 1.4% (Bindawas & Vennu, 2018).

Barriers to access and inclusivity of DHI to public and private life remain a challenge (Huyck et al., 2021). In Australia, language planning has been discussed to improve healthcare services to DHI (Huyck et al., 2021). High levels of deaf space and workplace inclusivity have been established (Wahat et al., 2023). For students who have DHI, understanding the challenges encountered by students with hearing impairment in higher education is paramount by recognizing the multifaceted nature of these challenges and exploring avenues for fostering an environment that not only accommodates but also empowers students with hearing impairment (Powell, Hyde & Punch, 2014).

Several international studies have been conducted to investigate the burden of deaf and hearing-impaired individuals (DHI). DHI patients were reported to be more likely to receive inaccessible communication, and accessing healthcare was considered to be unpleasant, frustrating, and time-consuming (James et al., 2022). Health-related quality of life (QoL) among DHI was shown to be significantly lower (Dalton et al., 2003). Hearing loss was also found to be substantially related with sadness and a lower quality of life (Huang et al., 2023). Focus groups with the deaf community revealed difficulty in getting health care information and services (Lieu et al., 2007). Some deaf patients expressed their belief that their doctors were culturally insensitive, stating that they frequently did not maintain face-to-face contact or correctly enunciate when speaking with deaf individuals (Ubido, Huntington & Warburton, 2002). Moreover, deaf patients believe that their medical professionals undervalue their intelligence, drive, and willingness to comprehend and take part in their care (Lieu et al., 2007). Conversely, nurses need to educate themselves on the distinctions between being deaf and belonging to the Deaf community (Lieu et al., 2007). It is not a given for nurses to presume that lip reading is the best way for a deaf patient to communicate (Bastable, 2017). In order to help the deaf patient who is trying to read lips, nurses should talk clearly and regularly, avoid screaming, and avoid over-enunciating to the point of distorting lips. When the discussion comes to a close, the nurse should make sure the patient is aware of the circumstances and make any required plans for a follow-up meeting (Barnett, 2002). In order to provide DHI patients with high-quality treatment, many nurses had no idea on how important it was for them to ensure and achieve a successful communication (Pendergrass et al., 2017). As a result, a large number of them rely on interpreters, friends, and relatives of DHI patients instead of speaking when they meet with them (Hosnjak et al., 2023). In addition, there is a deficiency in the necessary education, experience, and self-efficacy to interact with DHI patients in a successful manner (Velonaki et al., 2015). The experience of the healthcare provider affects both the standard of patient care and the practitioner. This effect is linked to various obstacles, including interpersonal, interactional, cultural, linguistic, and communicational ones, which lead to unequal access to healthcare for the deaf community and difficult circumstances for medical personnel (Ruiz-Arias & Barría-Pailaquilén, 2023). In a particular study, nurses felt that their lack of preparation in terms of deaf awareness and lack of training in sign language negatively impacts the quality of care they give to DHI individuals. In contrast, those who were somewhat prepared for this care often did so through attending particular courses and having personal experiences with deaf friends and family (Da Silva dos Santos et al., 2021).

The challenges in communication and caring for DHI patients has been highlighted from the healthcare staff but few solutions have been proposed and there is little evidence on staff perceptions of a potential intervention to address these issues. Because of this, the study was conducted to determine how Saudi health care providers’ (HCPs) perspectives on the use and effect of sign language (SL) knowledge on the provision of high-quality care to DHI patients.

Methods

This cross-sectional descriptive study was carried out in Saudi Arabia between October 2023 and January 2024 among healthcare professionals (HCPs) from various healthcare facilities who are actively serving in King Saud University hospital and four primary care clinics in Riyadh, Saudi Arabia. The Research Ethics Committee of the Deanship of Scientific Research at King Saud University in Riyadh, Saudi Arabia, gave the go-ahead to carry out the study with the approval document number KSU-HE-23-465.

The King Saud University Department of Surveys formally sent the survey to target HCPs via email and WhatsApp messaging platform. Prior to completing the survey, the participants were required to agree to participate and that was used as a written informed consent. The answers were downloaded as a Microsoft Excel worksheet. The data was gathered during a three-month period. It was made clear to participants that they were sampled but had the option to participate or not.

Demographic information (gender, age, clinical work experience, specialization, work area, educational level, and number of years since program completion) were among the data gathered. Questions on the importance of SL knowledge in communication with DHI patients, the effect of use of SL in effective communication with DHI patients, effect on the quality of health care delivery and improvement in the delivery of care with effective use of SL in communication were also asked. HCPs’ information were de-identified using serial numbers, and the final datasheet did not include any personal contact information like name, address, email address, and mobile number. The collected data were not distributed to any parties and was only utilized in this study.

Sample size was calculated using the formula n = Z2 × P × (1−P)/d2, where n is the sample size, Z is the level of confidence, P is the population and d is the margin of error. Using a 95% confidence level, at least 20% of the target population of HCPs in our institution, and a 5% margin of error, the calculated sample size was 246. From the final data, we grouped the respondents into two; group 1 for the nursing professionals and group 2 for other HCPs. Responses were compared between the two groups using the Chi-square test or Fisher’s exact test, whichever is applicable. Analysis of the data was performed using the Statistical Package for Social Sciences (SPSS) version 26.0 (IBM-SPSS, Armonk, New York, USA). A p value of <0.05 was considered statistically significant.

Results

A total of 238 HCPs were included in the study, 180 (75.6%) were nursing professionals and 58 were from other health specialties. Two hundred and nine (87.8%) were females and more than half (n = 130, 54.6%) were aged between 29 and 39 years old. Table 1 shows the detailed demographic characteristics of all HCPs.

Table 1 Demographic characteristics of 238 surveyed HCPs.

Characteristics	n (%)	
Gender		
Male	29 (12.2%)	
Female	209 (87.8%)	
Age groups in years		
18 to 28	27 (11.3%)	
29 to 39	130 (54.6%)	
40 and older	81 (34.0%)	
Clinical work experience in years		
<1	15 (6.3%)	
1 to 5	44 (18.5%)	
>5	179 (75.2%)	
Work area		
outpatient	80 (33.6%)	
inpatient	123 (51.7%)	
primary care center	35 (14.7%)	
Mother language		
Arabic	181 (76.1%)	
English	33 (13.9%)	
Other	24 (10.1)	
Educational levels	
Diploma	40 (16.8%)	
Bachelor	121 (50.8%)	
Masters	36 (15.1%)	
PhD	41 (17.2%)	
Number of years since completing education		
0–2	36 (15.1%)	
3–5	62 (26.1%)	
6–10	51 (21.4%)	
>10	89 (37.4%)	

There were 15 (6.3%) HCPs who claimed to have received formal training in SL. One hundred and sixty-five HCPs (69.3%) perceived that knowledge in SL is very important for communication with deaf and/or hearing-impaired (DHI) patients and their families, whereas 65 (27.3%) perceived SL as somewhat important (Fig. 1). One hundred and forty-one HCPs (59.2%) felt uncomfortable in communicating with DHI patients who use SL, while 96 (40.3%) preferred to ask family members or friends to interpret SL in communication. The most common barrier perceived by HCPs was the lack of resources (n = 119, 50.0%) and lack of time to learn SL (n = 51, 21.4%) (Fig. 2). One hundred and twenty-two HCPs (51.3%) claimed that SL interpretation is not available in their work setting, but 205 HCPs (86.1%) were interested to learn and receive training in SL.

Figure 1 Importance of knowledge in SL to communicate with deaf and hearing-impaired patients.

Figure 2 Perceived barriers to learning SL among HCPs.

Table 2 shows the comparative description of HCPs according to nursing professionals and other HCPs. Nursing professionals were significantly females (p < 0.001), were younger, aged between 29–39 years old (p < 0.001), were more assigned in the inpatient areas (p = 0.019), had more native English speakers (p < 0.001), and perceived the importance of SL in communication with DHI patients compared to other HCPs (p = 0.040). There were no significant differences with regards to clinical work experience (p = 0.114), receiving formal training in SL (p = 0.304) and frequency of encounter with a DHI patient (p = 0.364) between the two groups.

Table 2 The detailed description of HCPs according to nursing professionals and other HCPs.

Characteristics	Nursing
n = 180	other HCPs
n = 58	p values	
Gender, n (%)				
Male	15 (8.3%)	14 (24.1%)	0.001**	
Female	165 (91.7%)	44 (75.9%)	
Age groups in years, n (%)				
18–28	26 (14.4%)	1 (1.7%)		
29–39	105 (58.3%)	25 (43.1%)	<0.001**	
40 and older	49 (27.2)	32 (55.2%)		
Clinical work experience in years, n (%)				
<1	14 (7.8%)	1 (1.7%)		
2–5	36 (20.0%)	8 (13.8%)	0.114	
>5	130 (72.2%)	49 (84.5%)		
Work area, n (%)				
Outpatients	52 (28.9%)	28 (48.3%)		
Inpatient	98 (54.4%)	25 (43.1%)	0.019**	
Primary care center	30 (16.7%)	5 (8.6%)		
Mother language, n (%)				
Arabic	125 (69.4%)	56 (96.6%)		
English	31 (17.2%)	2 (3.4%)	<0.001**	
Others	24 (13.3%)	0		
Educational level, n (%)				
Diploma	38 (21.1%)	2 (3.4%)	<0.001**	
Bachelor	111 (61.7%)	10 (17.2%)	
Master	26 (14.4%)	10 (17.2%)	
PhD	5 (2.8%)	36 (62.1%)	
Notes.

** Significant.

Compared to other health care providers, nurses believed that the quality of health service and care for DHI patients is impacted by the inability of HCPs to communicate effectively and deliver high-quality care without the knowledge and understanding and use of SL (p = 0.016). Furthermore, nursing professionals believed that knowledge in SL improves the quality of care provided to DHI patients compared to other HCPs (97.2% vs. 87.9%, p = 0.005) (Fig. 3).

Figure 3 Effect of use of SL on quality of care and compromise DHI patients health care delivery.

Discussion

This study examined how SL affected the provision of high-quality healthcare to DHI patients from the viewpoints of allied health professionals, including nurses. Communication barriers between experts and people with hearing impairments is a major issue in the majority of public and private health services. The quality of care provided to the deaf community is severely compromised by this result, which maintains the barriers to communication between the sign and oral languages (Sheppard, 2014). Various medical professionals have varying effects and provide different levels of high-quality care (Abramowitz, Coté & Berry, 1987). Since nurses tend to the majority of patients and are the first-in-line at the hospital, their nursing care is seen to be the most crucial component in the provision of high-quality medical treatment (Edfort, 2024; Suchkov et al., 2024).

It is difficult to provide DHI patients with high-quality medical care (Mathew & Dannels, 2024; Rogers et al., 2024). With DHI patients, communication between nurses and other healthcare personnel becomes challenging and almost always results in lower-quality care, even in the presence of family and friends who can support the patient (Alamro et al., 2023). Furthermore, the same study found that a large number of nurses felt that their inability to successfully communicate with deaf and/or hearing-impaired (DHI) patients and provide high-quality care is hampered by their lack of knowledge, comprehension, and usage of SL, in comparison to other HCPs (Alamro et al., 2023). Even in hospitals and other healthcare facilities where interpreters are available, many nurses are unaware of the importance in ensuring efficient communication, which hinders the provision of high-quality care to DHI patients (McAleer, 2006).

DHI patients are frequently dissatisfied since they do not obtain enough information about preventative healthcare due to the significant barriers that they face on their own and the weak or limited understanding of nurses on SL (Pribanić & Milković, 2017). Because of the nurses’ lack of SL expertise which forms as a barrier to communication, over half of the nurses who responded in this study felt uneasy when speaking with DHI patients. This might get in the way of providing better care, support, or SL to DHI patients from the staff perspective, as well as workload, the division of patient care and the mental and emotional labor of additional SL translation work. Usually, the information is shared by nurses and healthcare providers with friends or family, but this becomes challenging when you take into account DHI patients’ rights to actively engage in the provision of high-quality care. In the end, every DHI patient is entitled to universal information about what is happening to their bodies and the choice to accept or refuse medical care (Park et al., 2024). Similar to the findings of this study, a study involving nurses from Fortaleza/Ceará hospitals reported challenges in communicating fully with deaf individuals. The study revealed that the professionals were uncertain about how to interact with deaf patients due to language barriers and their inability to gather crucial health information (Yet et al., 2022). Additionally, poor communication with DHI patients was a result of time restrictions brought on by the dearth of registered nurses, which has an impact on the quality of nursing service (Awases, Roos & Janetta, 2013). Additionally, a study revealed that a significant number of nurses believed that using family members as sign language interpreters had a detrimental impact on the privacy of DHI patients, especially when sensitive personal information was being requested (Steinberg et al., 2006).

While many of our respondents expressed interest in learning and receiving training in SL, most of our participants stated that SL interpretation and training are not available in their workplace. Institutional strategies and the provision of training for nursing staff may solve the shortage of learning resources and time for learning SL. Promoting education through simulation skills for health literacy and communication skills for DHI patients among nurses and nursing students is one suggested strategy (Whited et al., 2024). Another is the utilization of technology, such the Deaf in Touch Everywhere smartphone application, which is being utilized in Malaysia to meet the requirements of health care providers, SL interpreters, and the deaf community (Chong et al., 2024). Although research into these technology advancements and ongoing education aimed at fostering effective communication between medical professionals and the deaf community is ongoing, action must be taken to ensure that everyone receives the highest caliber of treatment. Terry and Meara proposed deaf awareness training during health professional training programs and provide basic deaf awareness information along with information about local Deaf communities and sign language training providers (Terry & Meara, 2024). eLearning deaf awareness programs was suggested as well to increase the knowledge and confidence of nurses when communicating with D/deaf and hard of hearing patients (Terry et al., 2025).

There are certain limitations to this study, despite the fact that it has explored how nurses and other healthcare professionals view the significance and influence of SL knowledge in interactions with DHI patients. One is the study’s survey design, which depended on self-reported data for responses. This kind of data is prone to biases due to a variety of factors, such as response variations and recollection, as well as the sample population’s dependability. The quantitative nature of the study could not gather reasons why staff found SL/training important or not important, or what factors might make them more or less likely to want or be able to learn and use SL with DHI patients. Second, in order to account for the impact of misunderstanding on the standard of care, patients’ compliance with management, and the provision of health services, we were unable to document the final result of miscommunication on the delivery of care to DHI patients. These study limitations suggest that future research using a qualitative or mixed methods approach is warranted, which might add further to the research evidence in this area.

Conclusion

In order to provide DHI patients with high-quality healthcare, nurses believe that understanding SL is essential. Nonetheless, relatively few nurses have formal training in SL and many have adequate knowledge of SL. In order to provide the highest caliber of care, it is necessary to improve communication with DHI patients through SL training. Healthcare institutions and health ministries should initiate and provide nurses and healthcare providers an area where they can potentially learn the art of SL for free. Delivery of optimal care to DHI patients and their inclusion to universal health provision is necessary, in line with the World Health Organization’s Universal Health Coverage and “health care for all”.

Supplemental Information

Supplemental Information 1 Raw data

Responses from the 238 participants including all types of the healthcare providers including physicians, pharmacists, pharmacy technicians, x-ray technicians. The majority were female nurses. Analysis of the data was performed using the Statistical Package for Social Sciences (SPSS) version 26.0 (IBM-SPSS, Armonk, New York, USA). Results are reported as numbers and percentages. To test the association between two categorical variables, we used the Chi-square test. A p value of <0.05 was considered statistically significant.

Supplemental Information 2 Study survey

There was no separate consent form provided to the participants, however, there was a statement at the top of the survey informs the participants about the nature of the study. They were informed that they could choose not to participate and that proceeding on the survey confirms their agreement to participate.

Additional Information and Declarations

Competing Interests

Author Contributions

Human Ethics

Data Availability

The author declares there are no competing interests.

Omaimah Qadhi conceived and designed the experiments, performed the experiments, analyzed the data, prepared figures and/or tables, authored or reviewed drafts of the article, and approved the final draft.

The following information was supplied relating to ethical approvals (i.e., approving body and any reference numbers):

The Research Ethics Committee of the Deanship of Scientific Research at King Saud University in Riyadh, Saudi Arabia (approval document number KSU-HE-23-465).

The following information was supplied regarding data availability:

The data are available in the Supplementary File.

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
