# Peer review of "Sign language use in healthcare: professionals’ insight"

_PeerJ, doi:10.7717/peerj.19446_

## Round 0.1 · original submission · Major Revisions

The authors are requested to carefully revise the manuscript and answer the questions raised by the reviewers.

·

Basic reporting

The introduction needs to be expanded, and needs to set the scene a bit better rather than jumping straight into the numbers of DHI people. Perhaps talk about how inclusivity and equality is crucial for effective healthcare. Then you can continue, as you have started, by indicating that there are problems with health inequality for people with disabilities, and in particular DHI people experience substantial barriers in accessing healthcare and poorer satisfaction with the care they do receive. I would recommend moving the statistics of how many people in SA are DHI, and how many worldwide (not the US as this is less relevant) earlier on, to keep the overview of existing literature about health equity barriers all together. Differentiate clearly about which studies in the introduction are from international settings, and which are specifically about SA DHI patients (group them together).

The language is mostly clear and professional, just needs a minor proof-read (e.g. throughout “majority” needs to be “A majority” or “The majority”). The structure is clear and robust, the data has been shared openly as have the questions. Two minor points: first, Tables 1-4 has only binary gender with no “other” or “prefer not to answer” option. I can appreciate that culturally perhaps binary gender is standard in this national setting, for the purposes of future meta-analysis the authors could add an “Other” or “prefer not to answer” option in the table, even if the figure is then 0.

Experimental design

The research topic fits the aims and scopes of PeerJ, and it is based on original empirical research.
At the end of the introduction there needs to be a stronger justification for this study; the current text has already highlighted the challenges of caring for DHI patients from the staff perspective, so is the justification that the challenges are apparent but few solutions have been proposed? Or a solution in the form of SL training has been proposed (in the literature, where?) but there is little evidence on staff perceptions of this potential intervention?

The methods are mostly well done. A few minor comments – first clarify was the survey opt-in, was it clear to the staff participants that they were sampled but able to select to consent to participate or not? Second, if significance is p=<0.05, please indicate within tables 2-4 with ** where any findings are significant.

Validity of the findings

Please note I am unable to comment on the validity of the statistical analysis as I am not a statistician.

The discussion is well done examining results and comparing with extant literature. One thing missing from the discussion is a critical think about what barriers might get in the way of providing better care, support or SL to DHI patients from the staff perspective; things like workload, the division of patient care labour between professions, and the mental and emotional labour of additional SL translation work. Solutions are offered, but it might be better to group these together somewhat in a “Practice and Policy Recommendation” session. Would also like to see a Limitations setting – for example the quantitative only study could not gather reasons why staff found SL/training important or not important, or what factors might make them more or less likely to want or be able to learn and use SL with DHI patients, suggesting that future research using a qualitative or mixed methods approach might add further to the research evidence in this area.

Additional comments

Thank you for the opportunity to review this study which presents findings from a survey of healthcare professionals in Saudi Arabia (SA) focusing on their experience with patients who are deaf or hearing impaired (DHI) and their perceptions of sign language training to help them care for these patients. This is a welcome addition to health inequalities literature and particularly useful in its focus on solutions to inequality on the basis of hearing-related disability. With some minor amendments it would make an excellent addition to the literature.

Reviewer 2 ·

Basic reporting

While English used is clear, sentence construction throughout manuscript needs correction- have indicated a few on the manuscript. Lit review is adequate. No hypothesis mentioned, the work does not have great novelty but may be of interest to the country where it was carried out.

Experimental design

This study is seen as a very basic quantitative study with very little scientific novelty.
The questionnaire used has not been validated, how participants were selected is also unclear. Raw data has ambiguous entry. Statistics used very very basic as well.
A thorough scientific examination is not observed.

Validity of the findings

Impact is only seen at country of research, perhaps some mention of Future work and recommendations in other scenario is warranted.

Additional comments

See comments in attached pdf

Annotated reviews are not available for download in order to protect the identity of reviewers who chose to remain anonymous.

·

Basic reporting

Thank you for the opportunity to review this interesting and worthwhile study. My comments mostly relate to minor tweaks regarding phrasing and wording.

I note this is a single author paper, yet the word 'we' appears early on, will you be including others as authors?

I'm thinking about the research question and the paper's title. Is it 'are they trained' or 'are they ready' as these are two different things, and many other elements as well in terms of competence and confidence. Maybe make sure you are 100% happy with the final wording.

Intro:
line 87, revise wording, as end of sentence is not clear - ending sentence ...'with 1.4%' is not clear to the reader.

line 90 - good opportunity to educate readers with more specific detail - what are difficulties for Deaf community getting health information, do you mean access issues, literacy issues, please explain a bit more.

line 96/97 - do you think your readers will know what this means, as if health professionals and the general public are mostly not aware, it may be best to explain.

line 113, instead of 'complete lack of preparation for this kind of care negatively' this may be better phrased as ...'lack of preparation in terms of Deaf awareness and lack of training in Sign language'. I can see that the topic is about people's experience of care, and the focus seems to be on communication (rather than 'this kind of care').

Line 116 - readers will wonder who 'we' are, the others involved, what were their backgrounds and ? research experience?

Material and methods:
line 122 - can you say 2 hospitals and 3 community settings, or something that is less vague?

line 126 - usually wording would 'provided ethical approval for the study to commence' or 'gave ethical agreement for the study to start'.

line 129 - where informed consent is concerned, I suggest this isn't an opportunity, this is absolutely required by the researcher for all participants. So you could say 'were required to provide consent......'


Results:
lines 158, and 167 would benefit from 'The...' at the start of the sentence.

Watch grammar and wording in places, e.g. line 186 'do not exist'

line 200 - when you say 'lack of resources' re Sign language learning, it would be good to be more specific here.

Discussion:
line 214 - plural, so apostrophe goes after the s - patients' primary language

line 231 - ? do they establish their own culture or are you recognizing that Deaf culture has already been established, and of course continues to develop and grow.

line 262 - what is institutionalised Sign language? do you mean that training is provided by or supported by the organisation?

line 278 - if the study was about the perception of Sign language, is that the same as readiness or ability or confidence? I'm taking you back to my earlier comments about the paper title. I think consistency of wording will be good.

Conclusion:
line 297 - where might staff potentially learn Sign language - I didn't see that in the paper anywhere and it's worth including that information.

If it's helpful, we have had some recently published studies this year about the lack of Deaf awareness training for health professionals, which does include their lack of knowledge of Sign language:

Terry, J., & Meara, R. (2024). A scoping review of Deaf awareness programs in Health professional education. PLOS Global Public Health, 4(8), e0002818.

Terry, J., Parkinson, R., Meara, R., England, R., Nosek, M., Humpreys, I., & Howells, A. (2024). Nursing students' knowledge of working with D/deaf and hard of hearing patients: Evaluation of a deaf awareness elearning package. Nurse Education Today, 106446.

Experimental design

Agree, this is original primary research relevant to the journal with a clear defined research question and relevant standards adhered to.

Validity of the findings

Agree findings have been presented and discussed in depth, with conclusions well stated.

Additional comments

No further comments.

---

## Round 0.2 · Minor Revisions

The authors are requested to carefully revise the manuscript and answer the questions raised by the reviewer.

·

Basic reporting

I note the importance of the topic, and my main comments are about further proofreading that is needs to address minor errors in the text. There are quite a few of these currently.

The paper itself is clear, but will benefit from corrections.

Experimental design

As per previous version, I have no issues regarding the design and implementation of the study.

Validity of the findings

The findings are reported clearly. I am not a statistician.

Additional comments

Thank you for the opportunity to review this re-submission. I can see that care and attention have paid to comments from previous reviewers. My comments are brief, but will help the readability of the paper.

Line 32: add ‘…and may be..’ before social withdrawn
Line 32: ‘often may have emotional.’
Line 33: ‘face’ present tense
Line 35: does not make sense, did you mean ‘hearing loss’ as it looks as if you are talking about hearing people.
Line 79 has extra full stop

I have just pointed out some of the typos, missing words and things that can be picked up with careful proofreading. I am seeing more on the following pages, Please can all pages be checked very thoroughly.

Line 137/138 needs revising as meaning is vague.
References need some re-aligning in terms of indentation and font to achieve consistency. However, do please be consistent regarding journal’s referencing system, as a variety of formats/versions are currently used.

---

## Round 0.3 · accepted · Accept

All reviewers left with minor revision. After revisions I think the author has responded adequately. I also reviewed the manuscript and found no obvious risks to publication. Therefore, I also approved the publication of this manuscript.